# Shared Risk Factors between Dementia and Atherosclerotic Cardiovascular Disease

**DOI:** 10.3390/ijms23179777

**Published:** 2022-08-29

**Authors:** Liv Tybjærg Nordestgaard, Mette Christoffersen, Ruth Frikke-Schmidt

**Affiliations:** 1Department of Clinical Biochemistry, Copenhagen University Hospital—Rigshospitalet, 2100 Copenhagen, Denmark; 2Department of Clinical Medicine, Faculty of Health and Medical Sciences, University of Copenhagen, 2200 Copenhagen, Denmark

**Keywords:** Alzheimer’s disease, vascular dementia, body mass index, lipids, lipoproteins, hypertension, diabetes, physical inactivity, smoking, diet

## Abstract

Alzheimer’s disease is the most common form of dementia, and the prodromal phases of Alzheimer’s disease can last for decades. Vascular dementia is the second most common form of dementia and is distinguished from Alzheimer’s disease by evidence of previous stroke or hemorrhage and current cerebrovascular disease. A compiled group of vascular-related dementias (vascular dementia and unspecified dementia) is often referred to as non-Alzheimer dementia. Recent evidence indicates that preventing dementia by lifestyle interventions early in life with a focus on reducing cardiovascular risk factors is a promising strategy for reducing future risk. Approximately 40% of dementia cases is estimated to be preventable by targeting modifiable, primarily cardiovascular risk factors. The aim of this review is to describe the association between risk factors for atherosclerotic cardiovascular disease and the risk of Alzheimer’s disease and non-Alzheimer dementia by providing an overview of the current evidence and to shed light on possible shared pathogenic pathways between dementia and cardiovascular disease. The included risk factors are body mass index (BMI); plasma triglyceride-, high-density lipoprotein (HDL) cholesterol-, low-density lipoprotein (LDL) cholesterol-, and total cholesterol concentrations; hypertension; diabetes; non-alcoholic fatty liver disease (NAFLD); physical inactivity; smoking; diet; the gut microbiome; and genetics. Furthermore, we aim to disentangle the difference between associations of risk factors in midlife as compared with in late life.

## 1. Introduction

Dementia is a clinical diagnosis defined by cognitive symptoms that interfere with the ability to function at usual activities [1]. It is a devastating neurodegenerative disease affecting more than 50 million individuals worldwide [2]. The most common symptoms are problems with memory, language, problem solving, and other thinking skills [3], and it is preceded by a period during which individuals have subtle cognitive changes called mild cognitive impairment (MCI) [3]. The dementia syndrome is categorized as mild, moderate, or severe [3].

Alzheimer’s disease is the most common form of dementia, and the prodromal phases of Alzheimer’s disease can last for decades [4]. The disease is categorized as early-onset (before age 65) and late-onset (after age 65) dementia [5]. It has been viewed as a mysterious disease with no single obvious cause, different hypotheses regarding its pathogenesis, and no curative treatment. Even though β-amyloid is the pathological hallmark of Alzheimer’s disease, the amyloid cascade hypothesis that states Alzheimer’s disease is mainly caused by the accumulation of toxic β-amyloid [6] has been challenged several times. Clinical randomized trials aimed at reducing the β-amyloid burden have failed to show any effect on disease endpoint [7,8,9,10,11], and β-amyloid plaques have been shown to be present in cognitively normal individuals [12] and individuals without dementia [13].

Vascular dementia is the second most common form of dementia and is distinguished from Alzheimer’s disease by evidence of previous stroke or hemorrhage and current cerebrovascular disease [3]. Other types of dementia include frontotemporal [14], Lewy body dementia [15], unspecified dementia [3], and mixed dementia [3]. Many of the types of dementia in the unspecified dementia category are likely to be of vascular origin, and therefore, unspecified dementia and vascular dementia are often categorized as non-Alzheimer dementia [16,17]. This classification will be used throughout the review.

Recent evidence indicates that curing or reverting Alzheimer’s disease in the prodromal phases or after disease onset is futile. In contrast, preventing the disease by lifestyle interventions early in life with a focus on reducing cardiovascular risk factors is a promising strategy for reducing the future risk of dementia [16,18]. One study estimated that approximately 40% of dementia cases worldwide could be prevented by targeting modifiable, primarily cardiovascular risk factors [2]. Furthermore, neuroimaging studies have suggested that early vascular dysregulation is the initial pathologic event leading to late-onset Alzheimer’s disease [19,20].

The aim of this review is to describe the associations between risk factors for atherosclerotic cardiovascular disease including body mass index (BMI); plasma triglyceride-, high-density lipoprotein (HDL) cholesterol-, low-density lipoprotein (LDL) cholesterol-, and total cholesterol concentrations; hypertension; diabetes; non-alcoholic fatty liver disease (NAFLD); physical inactivity; smoking; diet; the gut microbiome; genetics; and risk of dementia. We aim to give an overview of the current evidence and to shed light upon possible shared pathogenic pathways between atherosclerotic cardiovascular disease and Alzheimer’s disease, non-Alzheimer dementia, and all-cause dementia. Furthermore, we aim to disentangle the difference between associations of risk factors in midlife compared with in late life.

## 2. Alzheimer’s Disease and Vascular Dementia

General diagnostic criteria for dementia according to the International Classification of Diseases (ICD) version 10 include: (1) decline in memory and decline in other cognitive functions, (2) preserved consciousness sufficient to be able to judge decline in memory, (3) weakening of emotional control, motivation, or social interaction, (4) duration of more than 6 months. In addition, the symptoms must not be explained by other major psychiatric disorders [3]. Dementia has many causes, and Alzheimer’s disease and vascular dementia are the two most common. A diagnosis of either form of dementia is based on family and medical history, cognitive tests such as the Mini-Mental State Examination (MMSE), and input from close relatives about the patient’s changes in behavior, abilities, and/or mood [3,4].

Alzheimer’s disease was first described in 1907 by Alois Alzheimer [21]. The post-mortem examination of the patient’s brain was described with arteriosclerotic changes in the vascular tissues and thick bundles of neurofibrils that stained with dyes that normally would not stain neurofibrils (later known as neurofibrillary tangles) [21]. Biomarkers can be used to distinguish Alzheimer’s disease from other types of dementia [4]. These include early signs of Alzheimer’s disease such as abnormal levels of β-amyloid shown either by examination of cerebrospinal fluid or in positron emission tomography (PET) scans and decreased glucose metabolism shown on PET scans [4].

Vascular dementia has different causes which can be classified according to the following [22]: multiple infarcts, strategic single-infarct, small-vessel disease with dementia, hypoperfusion, hemorrhagic dementia, and other causes [22]. Alzheimer’s disease and vascular dementia share many clinical features and their differential diagnosis requires brain imaging and or/postmortem assessment [23]. The pathological characteristics of vascular dementia are microinfarcts, lacunar infarcts, arteriosclerosis, lesions caused by hemorrhages, fibroid necrosis, and hyalinosis [23]. While all these features can also be seen in Alzheimer’s disease, the pathological hallmark of Alzheimer’s disease is the accumulation of β-amyloid [23,24]. When β-amyloid accumulates in the capillaries of the brain it causes cerebral amyloid angiopathy, which is very often seen in Alzheimer’s disease and often causes cerebral hemorrhages [24]. Cerebral amyloid angiopathy is also present in vascular dementia, however, amyloid plaques are rarely seen [24].

## 3. Shared Risk Factors between Dementia and Atherosclerotic Cardiovascular Disease

Recently, an increasing amount of evidence points towards risk factors for atherosclerotic cardiovascular disease as risk factors for dementia [16]. Diabetes mellitus type 2 and midlife hypertension have been robustly associated with risk of dementia, whereas other risk factors such as obesity, plasma concentrations of LDL cholesterol, HDL cholesterol, triglycerides and total cholesterol, NAFLD, diet, the gut microbiome, smoking, and physical inactivity are less well-established risk factors [16]. Increasing evidence also suggests that genetic factors may be shared risk factors for dementia and atherosclerotic cardiovascular disease [16,17,18]. The associations between risk factors and dementia vary according to what time point in life they have been measured and which type of dementia is the outcome. This often complicates the interpretation of research within this field. An overview of shared risk factors between dementia and atherosclerotic cardiovascular disease, discussed in this review, is given in Figure 1.

## 4. Body Mass Index

### 4.1. Alzheimer’s Disease

In a meta-analysis from 2019 conducted by Lee et al. that included 2.8 million individuals and 57,000 dementia cases, the observational association between BMI categories and a risk of non-vascular dementia (which includes Alzheimer’s disease) was U-shaped with the lowest risk in the upper normal weight BMI category (22.5 to 24.9 kg/m^2^) versus the lower normal weight (18.5–22.4 kg/m^2^) category [25]. The highest risk was seen in the underweight BMI category (<18.5 kg/m^2^). In a Mendelian randomization study that included approximately 54,000 individuals, Østergaard et al. observed no association between genetically predicted BMI and a risk of Alzheimer’s disease [26]. This was in line with the results from the study by Nordestgaard et al. where a clear observational association between low BMI and a high risk of Alzheimer’s disease was reported but no association between genetically determined BMI and a risk of Alzheimer’s disease [27]. Nearly 400,000 individuals were included in this Mendelian randomization study.

One likely explanation for the association between low BMI and a high risk of Alzheimer’s disease is reverse causation. Individuals in prodromal phases of dementia are known to eat less than before a dementia diagnosis and often experience weight loss [28,29]. There are different reasons for this weight loss that include changes in eating habits and food preference, a change in appetite, and swallowing problems [28]. Even in healthy elderly individuals, weight loss, termed “anorexia of aging”, is a known phenomenon mainly caused by reduced hunger resulting from lower energy requirements and more potent inhibitory satiety signals [30].

### 4.2. Non-Alzheimer Dementia

Lee et al. observed an increased risk of vascular dementia in the underweight BMI category as well as in the obese BMI category (≥30 kg/m^2^). To minimize the risk of reverse causation, the longest available left-censored data were also used which increased the association between obesity and high risk of vascular dementia [25]. Obesity influences several risk factors for atherosclerotic cardiovascular disease including plasma triglycerides and LDL cholesterol concentrations, and diabetes [31]. The development of atherosclerosis could lead to strokes which further could cause non-Alzheimer dementia (see Figure 2 and Figure 3).

### 4.3. All-Cause Dementia

In 2009, Fitzpatrick et al. showed that midlife (measured at age 50) obesity was associated with a high risk of developing all-cause dementia compared with underweight, normal weight, and overweight BMI categories. However, when looking at BMI at age 65 or older, the highest risk of developing all-cause dementia was in the underweight category compared with the other BMI categories [32]. In a large retrospective cohort study, Qizilbash et al. investigated the observational association between body mass index and a risk of all-cause dementia [33]. They included almost 2 million people with a median age at baseline of 55 years and found that having a BMI below 20 was associated with a high risk of all-cause dementia. They concluded that their results contradicted the hypothesis that high BMI in midlife was a risk factor for developing all-cause dementia [33].

However, in 2019, Lee et al. conducted a meta-analysis that included 2.8 million individuals and 57,000 all-cause dementia cases which lead to a different conclusion [25]. Here, they found that the observational association between BMI categories and a risk of all-cause dementia was U-shaped with the lowest risk in the upper normal weight BMI category (22.5 to 24.9 kg/m^2^) when using the lower normal weight (18.5–22.4 kg/m^2^) category as a reference [25]. The highest risk was seen in the underweight BMI category (<18.5 kg/m^2^). Singh-Manoux et al. found that a relatively high body mass at age 50 and a relatively low body mass index at age 70 was associated with a high risk of all-cause dementia, supporting the hypothesis that high BMI in midlife is a risk factor for developing all-cause dementia [34]. However, when using a Mendelian randomization approach on approximately 40,000 individuals and including a polygenic score strongly associated with BMI, Mukherjee et al. found that genetically determined BMI was not associated with a risk of all-cause dementia [35]. 

The association between midlife obesity and the risk of all-cause dementia may have different explanations. The effect of midlife obesity could be mediated through hypertension since obesity has been shown to be causally associated with hypertension [36], and hypertension is a risk factor for all-cause dementia [2].

## 5. Lipids and Lipoproteins

The two most abundant lipids in the plasma compartment are total cholesterol and triglycerides. Since lipids are not soluble in the water phase of plasma, they are carried in lipoproteins which are lipid particles in association with proteins (See Figure 2). Lipoproteins include HDLs that are the smallest lipoproteins, LDLs that are medium-sized lipoproteins, triglyceride-rich lipoproteins (chylomicrons, very-low-density lipoprotein (VLDL), and intermediate-density lipoproteins (IDL)) that are the largest lipoprotein particles [37].

The cholesterol content in these lipoprotein particles is reported as HDL cholesterol, LDL cholesterol, and remnant cholesterol. Remnant cholesterol is the cholesterol content of all triglyceride-rich lipoproteins, and plasma triglycerides represent a marker for remnant cholesterol [37].

### 5.1. Plasma Triglyceride Concentrations

#### 5.1.1. Alzheimer’s Disease

Studies on plasma triglyceride concentrations and risk of Alzheimer’s disease are few and inconclusive [38]. In a study by Nordestgaard et al., they found that only low plasma triglyceride concentrations were associated with a high risk of Alzheimer’s disease [17]. Bearing in mind the observational association between low body mass index and high risk of Alzheimer’s disease [25,27], this association may be explained by reverse causation because a low body mass index is associated with low triglyceride concentrations, and the body mass index decreases in the prodromal phases of Alzheimer’s disease. Supporting this hypothesis, when adjusting for diabetes and body mass index, the association between low plasma triglyceride concentration (median 89 mg/dL) and a high risk of Alzheimer’s disease was attenuated [17]. Raffaitin et al. found that there was no association between high plasma triglycerides and a risk of Alzheimer’s disease, and when adjusting for other components of the metabolic syndrome, the association was even less significant [39]. Similar results were observed by Reitz et al. [40] and Wang et al. [41]. 

In conclusion, the association between low plasma triglyceride concentrations and a high risk of Alzheimer’s disease is most likely to reflect the association between low body mass index and a high risk of Alzheimer’s disease.

#### 5.1.2. Non-Alzheimer Dementia

In a meta-analysis from 2016, Anstey et al. reviewed the existing literature and found that there was no association between triglyceride concentrations measured in late life and a risk of vascular dementia [38]. In a combined cross-sectional (*n* = 2820) and prospective study (*n* = 1168) with individuals 65 years and older, Reitz et al. found that there was no association between plasma triglyceride concentrations and risk of incident vascular dementia, but a tendency towards an increased risk [40]. However, this analysis was based on 54 vascular dementia cases and 856 controls [40]. Raffaitin et al. found that high plasma triglyceride concentrations (above 150 mg/dL) were associated with a higher risk of incident vascular dementia in a sample of 7077 individuals [39]. Nordestgaard et al. found that plasma triglyceride concentration was associated with a risk of non-Alzheimer dementia (including vascular dementia and unspecified dementia), both on a continuous scale and when comparing the group with the highest plasma triglyceride concentration (above 629 mg/dL) to the group with the lowest plasma triglyceride concentration (below 89 mg/dL) [17].

The mechanism(s) behind the association between high plasma triglycerides and risk of dementia are unknown. If the association is assumed to be causal, it is likely to be on the causal pathway from diabetes and/or high body mass index to high plasma triglyceride concentration to high risk of ischemic stroke and non-Alzheimer dementia (see Figure 3). This pathway is plausible since (1) the most frequent causes of high plasma triglyceride concentrations are dysregulated diabetes mellitus and obesity [42,43]; (2) a diagnosis of vascular dementia requires, in addition to the general criteria of a dementia diagnosis, clear evidence of cerebrovascular disease (mainly ischemic stroke and/or hemorrhage); and (3) the association between plasma triglyceride concentrations and risk of ischemic stroke has been established previously [44].

In support of this, Nordestgaard et al. found that the associations between triglyceride concentrations and a risk of ischemic stroke as well as non-Alzheimer dementia were attenuated when adjusting for body mass index and diabetes mellitus status [17]. Raffaitin et al. found that the association between high plasma triglycerides and a risk of vascular dementia was attenuated when adjusting for other components of the metabolic syndrome (high blood pressure, high waist circumference, low HDL cholesterol, and high glycemia) [39].

High plasma triglycerides reflect a high cholesterol content of triglyceride-rich lipoproteins (see Figure 2), which are causally associated with atherosclerotic cardiovascular disease due to their cholesterol content [37,45,46,47,48,49,50,51,52,53]. Since the smallest triglyceride-rich particles can penetrate the arterial wall, the cholesterol content of these particles can get trapped in the arterial intima and lead to the development of atherosclerosis [53,54,55]. In addition, elevated levels of the cholesterol content in triglyceride-rich lipoproteins, which are highly correlated with levels of triglycerides, has been shown to cause low-grade inflammation [51] contributing to the development of atherosclerosis. Perhaps, the free fatty acids from hydrolysis of the triglyceride content of triglyceride-rich lipoproteins also contributes to the development of atherosclerosis [37]. If the association between high levels of triglyceride-rich lipoproteins marked by plasma triglycerides and a risk of non-Alzheimer dementia is causal, it is most likely mediated through ischemic strokes caused by atherosclerosis (see Figure 3).

#### 5.1.3. All-Cause Dementia

Raffaitin et al. found that high plasma triglyceride concentrations (above 150 mg/dL) were associated with a higher risk of incident all-cause dementia [39]. The metabolic syndrome, which includes high plasma triglycerides and inflammation, has also been associated with high risk of cognitive decline in a prospective study by Yaffe et al. including 2632 individuals with a mean age of 74 years [56]. As for non-Alzheimer dementia, this could reflect a causal role of triglyceride-rich lipoproteins in the development of atherosclerosis which is likely to be involved in the pathogenesis of all-cause dementia (see Figure 3).

### 5.2. Plasma High-Density Lipoprotein Cholesterol Concentrations

#### 5.2.1. Alzheimer’s Disease

In a population-based autopsy study of 218 Japanese American men with an average age at death of 85 years, high concentrations of HDL cholesterol measured in late life were associated with accumulation of more neurofibrillary tangles and neuritic plaques in neocortex and/or hippocampus, and midlife high HDL cholesterol concentrations were associated with more accumulation of neurofibrillary tangles in neocortex [57]. Late life was determined as 20 years after midlife, and midlife was defined as age 40–65 years. In a study by Retiz et al., they found that there was no association between high plasma HDL concentrations and a risk of Alzheimer’s disease. Recently, Kjeldsen et al. found that higher plasma HDL concentrations were associated with a higher risk of Alzheimer’s disease in a prospective cohort that included 112,000 participants with a median age of 57 and a mean follow-up of 10 years [58].

Variations in lipid metabolism genes that have been associated with plasma HDL cholesterol concentrations as well as a risk of Alzheimer’s include the apolipoprotein E gene (*APOE*) and the ATP-binding cassette transporter A1 gene (*ABCA1*) [59,60,61]. The *APOE* ɛ4 allele that is associated with high risk of Alzheimer’s disease is also associated with low plasma HDL cholesterol concentrations [62]. A loss-of-function mutation in *ABCA1* that has been associated with low concentrations of plasma HDL cholesterol has also been found to increase the risk of incident Alzheimer’s disease [61].

#### 5.2.2. Non-Alzheimer Dementia

Reitz et al. found that higher plasma HDL cholesterol concentrations were associated with a lower risk of prevalent but not incident vascular dementia [40]. A recent study on genetic variation in the cholesteryl ester transfer protein gene (*CETP*) found that high plasma HDL cholesterol concentrations were associated with a low risk of vascular dementia [63]. Since the relationship between plasma HDL cholesterol and triglyceride concentrations are usually inverse, these results support the hypothesis of high plasma triglycerides marking triglyceride-rich lipoproteins with high cholesterol content, as a causal risk factor for non-Alzheimer dementia. However, the role of plasma HDL cholesterol in relation to non-Alzheimer dementia risk remains unclear. Genetically determined HDL cholesterol has been shown to be not associated with development of atherosclerosis, likely excluding a causal role for HDL cholesterol through atherosclerosis [64].

#### 5.2.3. All-Cause Dementia

The results from observational studies on the association between plasma HDL cholesterol and a risk of all-cause dementia are conflicting [65]. In a cross-sectional analysis from 1996 that included 37 all-cause dementia cases and 297 controls, Bonarek et al. found that high plasma HDL concentrations were associated with a low risk of prevalent dementia [66]. In contrast, An et al. found high plasma HDL cholesterol concentrations to be associated with a high risk of all-cause dementia in a prospective study that included 2500 participants followed for two years with a median age of 59 years. Kjeldsen et al. found higher plasma HDL concentrations to be associated with a higher risk of all-cause dementia in a prospective cohort that included 112,000 participants [58].

Plasma HDL cholesterol concentrations have been shown to be associated with alcohol consumption, levels of inflammation [67], plasma triglyceride concentrations [67], and abdominal obesity [56], which are all risk factors for all-cause dementia. Thus, the association between high plasma HDL cholesterol concentrations and high risk of all-cause dementia might be the result of reverse causation due to low body mass index and low triglycerides, or a reflection of a high alcohol consumption, or both.

### 5.3. Plasma Total Cholesterol Concentrations

#### 5.3.1. Alzheimer’s Disease

In a meta-analysis from 2017 that included 23,000 individuals, Anstey et al. found that high total plasma cholesterol measured in midlife but not in late life was associated with a high risk of Alzheimer’s disease [38]. Reitz et al. found that high total cholesterol concentrations were associated with a low risk of incident Alzheimer’s disease when adjusting for sex, age, race, education, body mass index, apolipoprotein E genotype, diabetes, heart disease, and hypertension [40]. However, the mean age of the Alzheimer’s disease cases was 81 years, and the mean age of the control subjects was 78 years, thus, reflecting associations in late life [40]. Similar results were found in a later study by the same authors [68]. In a study by Launer et al., no association between midlife or late life total cholesterol concentrations and the amount of accumulation of neuritic plaques, senile plaques, or neurofibrillary tangles was found [57].

From this evidence, it seems as though high plasma total cholesterol measured in midlife is associated with a high risk of Alzheimer’s disease and high plasma total cholesterol measured in late life is associated with a low risk of Alzheimer’s disease. High total cholesterol in midlife is a risk factor for developing atherosclerosis in all arteries, including the arteries in the brain. High plasma total cholesterol measured in late life most likely reflects a high body mass index which has been shown to be associated with a low risk of Alzheimer’s disease most probably due to reverse causation. Thus, results on the association between plasma total cholesterol concentrations and a risk of Alzheimer’s disease support previous findings on the associations between body mass index and plasma triglyceride concentrations and a risk of Alzheimer’s disease.

#### 5.3.2. Non-Alzheimer Dementia

The literature on total plasma cholesterol concentrations and a risk of non-Alzheimer dementia is sparse. However, in a study by Reitz et al., they found that there was no association between high plasma total cholesterol concentrations and a risk of incident vascular dementia [40].

### 5.4. Plasma Low-Density Lipoprotein Cholesterol Concentrations and Use of Lipid-Lowering Therapy

#### 5.4.1. Alzheimer’s Disease

Reitz et al. found neither incident nor prevalent Alzheimer’s disease to be associated with levels of plasma LDL cholesterol concentrations [40]. In a study by Launer et al., LDL cholesterol concentrations measured in late life were not associated with accumulation of senile plaques, neuritic plaques, or neurofibrillary tangles [57]. In a large Mendelian randomization study that included 111,000 individuals, Benn et al. found that there was no observational association between plasma LDL cholesterol concentrations and a risk of Alzheimer’s disease [69]. They used a gene score created with proprotein convertase subtilisin/kexin type 9 (*PCSK9*) and 3-hydroxy-3-methyl-glutaryl-coenzyme A reductase (*HMGCR*) (the target of statins) variants. This score was associated with genetically determined low LDL cholesterol but not with Alzheimer’s disease [69].

Whether lipid-lowering therapy can be used in the prevention of Alzheimer’s disease is unclear. Reitz et al. found that the use of lipid-lowering therapy was associated with a lower risk of prevalent but not incident Alzheimer’s disease [40]. However, in this study in the analysis of incident disease, the mean age was 78 years and the mean age at onset for Alzheimer’s disease was 83 years. This suggests that the mean age was too high to allow for any final conclusions to be drawn. A meta-analysis by Chu et al from 2018 that included 16 studies found that the use of statins was associated with a relative risk of 0.72 (0.58–0.90) for Alzheimer’s disease [70].

#### 5.4.2. Non-Alzheimer Dementia

The evidence on the association between plasma LDL cholesterol and a risk of non-Alzheimer dementia is conflicting [16]. Reitz et al. found that higher plasma LDL cholesterol concentrations were associated with a high risk of incident vascular dementia but not with prevalent vascular dementia [40]. In a large Mendelian randomization study by Benn et al., they found that genetically determined low plasma LDL cholesterol concentrations were associated with a low risk of vascular dementia. When using summary risk estimates, only low LDL cholesterol concentration caused by *HMGCR* alleles seemed to be associated with a low risk of vascular dementia [69]. In the meta-analysis by Chu et al. there was no association between statin use and risk of vascular dementia [70]. Accordingly, Reitz et al. found no association between the use of lipid-lowering therapy and risk of prevalent or incident vascular dementia [40].

LDL cholesterol is known to cause atherosclerotic cardiovascular disease [71,72]. The most plausible explanation for the association between LDL cholesterol and risk of vascular dementia is, thus, through atherosclerosis. The fact that genetic variants associated with LDL cholesterol concentrations are also associated with risk of vascular dementia further supports this hypothesis.

#### 5.4.3. All-Cause Dementia

In a study by Benn et al., they found that there was no observational association between LDL cholesterol concentrations and a risk of all-cause dementia [69]. A Cochrane review from 2016 found that statins given in late life to people at risk of vascular disease did not prevent cognitive decline or all-cause dementia [73]. However, this review included only two studies (*n* = 26,000) with 3 and 5 years of follow-up [73], which was far from long enough time to develop all-cause dementia [74], which is a disease that often develops over decades [75]. In a meta-analysis from 2017 that included 31 studies, more than 3 million individuals and approximately 184,000 incident dementia cases statin use was associated with a relative risk (95% confidence interval) of 0.85 (0.80–0.89) for all-cause dementia. They found a 20% lower risk of all-cause dementia per one year of statin use [76]. In the meta-analysis by Chu et al. statin use was associated with a relative risk (95% confidence interval) of 0.85 (0.79–0.92) for all-cause dementia [70].

## 6. Hypertension and Antihypertensive Drugs

### 6.1. Alzheimer’s Disease

In a large meta-analysis by Tully et al., a reduced risk of Alzheimer’s disease was found for those taking diuretic antihypertensive medication compared with those not taking it [77]. In a study by Ding et al.the use of antihypertensive medication was associated with a lower risk of Alzheimer’s disease in those with high baseline systolic (≥140 mmHg) or diastolic (≥90 mmHg) blood pressure [78]. A possible causal role for hypertension in the development of Alzheimer’s disease could be through the induction of atherosclerosis in cerebral arteries and by causing micro- and macro bleeds, white matter lesions, and microinfarcts [79].Thus, hypertension could be an important and treatable risk factor for Alzheimer’s disease.

### 6.2. All-Cause Dementia

Most evidence points towards midlife hypertension as a risk factor for developing all-cause dementia, and antihypertensive drugs as a possible prevention strategy [2]. In a study that included 1440 individuals who were followed for approximately 26 years, McGrath et al. found that midlife hypertension was associated with a high risk of all-cause dementia [80]. However, among normotensives, they also found that steep declines in blood pressure in midlife and late life were associated with a high risk of all-cause dementia [80]. Likewise, Abell et al. found that systolic blood pressure above 130 mmHg at age 50 was associated with a higher risk of all-cause dementia, in a study that included 1462 women followed for 44 years [81].

Four different meta-analyses have also reported that antihypertensive medication was associated with a lower risk of all-cause dementia [82]. The first study by Tully et al. included 13 different cohorts and a total number of individuals of 52,599 with 3444 all-cause dementia cases, a median age of 76 years, and a median follow-up of 6 years [77]. They found diuretics to be associated with a reduced risk of all-cause dementia [77]. The second study that focused on calcium channel blockers and included 10 prospective studies totaling 75,000 individuals with a median age of 72 years and 8 years of follow-up, found a 30% reduced risk of all-cause dementia in calcium channel blocker users as compared with nonusers [83]. The third meta-analysis of four studies with more than 10 mmHg between intervention and the control group showed a hazard ratio of 0.88 (0.78–0.88) for all-cause dementia [82]. The fourth study by Ding et al. included six prospective studies and approximately 31,000 individuals and 3700 incident all-cause dementia cases. The mean age was 59 to 77, and the median follow-up was 7 to 22 years. In this study the use of antihypertensive medication was associated with a lower risk of all-cause dementia in those with high baseline systolic (≥140 mmHg) or diastolic (≥90 mmHg) blood pressure [78].

Blood pressure has been shown to decrease in the years before death [84]. A likely explanation for the differing associations across the lifespan is that hypertension, as a cardiovascular and cerebrovascular risk factor in midlife, can cause damage to the brain that will later result in development of dementia. In support of this hypothesis, midlife hypertension (from age 40 years) has been shown to be associated with reduced brain volumes, which is a common characteristic for brains affected by dementia [85]. The most likely explanation for the association between a decline in blood pressure in late life and a risk of dementia is, like the explanation provided for body mass index and plasma triglyceride concentrations, reverse causation.

In a study by the SPRINT MIND investigators (which included approximately 9300 individuals with hypertension aged 50 years or older and followed for 8 years) intensive blood pressure control (systolic blood pressure ≤120 mmHg) as compared with standard treatment (systolic blood pressure ≤140 mmHg) was associated with a lower risk of mild cognitive impairment but not with probable all-cause dementia [86]. 

Taken together, hypertension also seems to be an important and treatable risk factor for all-cause dementia.

## 7. Diabetes

### 7.1. Alzheimer’s Disease

In a large Mendelian randomization study from 2020 that included data on 200,000 individuals from a genetic consortia and observational data on 800,000 individuals from nationwide registers, Thomassen et al. found that a diagnosis of type 2 diabetes was associated observationally with a higher risk of Alzheimer’s disease [87]. They also used genetic instruments to determine whether the association between diabetes and risk of Alzheimer’s disease was causal and they reported an odds ratio (95% confidence intervals) of 1.04 (0.98–1.10) for those with diabetes compared with those without diabetes [87]. In a pooled analysis from 2016, Chatterjee et al. included 14 studies comprising 2.3 million people and more than 100,000 dementia cases to investigate the prospective association between diabetes and dementia. Relative risks for non-vascular dementia (including Alzheimer’s disease) were close to 1.5 for both men and women [88].

### 7.2. Non-Alzheimer Dementia

In a study by Thomassen et al., a diagnosis of diabetes was associated observationally with multifactorially adjusted hazard ratios (95% confidence intervals) of 1.98 (1.83–2.14) for vascular dementia and 1.53 (1.48–1.59) for unspecified dementia [87]. In a study by Chatterjee et al., relative risks (95% CI) for vascular dementia were 2.34 (1.86–2.94) in women and 1.73 (1.61–1.85) in men for individuals with diabetes compared with individuals without diabetes [88].

### 7.3. All-Cause Dementia

Overall, the current evidence points towards diabetes as a causal risk factor for developing all-cause dementia. In a study by Chatterjee et al. diabetes was associated with a 60% increase in risk of all-cause dementia for both men and women [88]. In a large, prospective population-based cohort study from 2015 that included 2.5 million individuals of which 223,000 had diabetes and 60,000 developed all-cause dementia, they found a hazard ratio of 1.20 (95% CI 1.17–1.23) for developing all-cause dementia for those with diabetes compared with those not having diabetes [89]. In a study by Thomassen et al. a diagnosis of type 2 diabetes was associated observationally with a hazard ratio (95% confidence intervals) of 1.48 (1.44–1.53) for all-cause dementia [87]. In conclusion, many large, observational studies and Mendelian randomization studies point in the same direction regarding the association between diabetes and risk of all-cause dementia. However, the underlying pathogenesis remains unknown.

## 8. Non-Alcoholic Fatty Liver Disease

### 8.1. Non-Alzheimer Dementia

NAFLD is the most common cause of chronic liver disease in Western countries [90]. Many risk factors have been associated with cardiovascular disease [91], but only a few studies have investigated the association between NAFLD and a risk of dementia. In a study from 2021 by Labenz et al. that included approximately 22,000 NAFLD cases and controls, no association between NAFLD and a risk of vascular dementia was found [92]. However, in a population-based cohort study from 2022 by Shang et al. that included 2898 NAFLD cases and 28,357 controls, NAFLD was associated with a hazard ratio (95% confidence interval) of 1.44 (0.96–2.23) for developing vascular dementia [93].

### 8.2. All-Cause Dementia

In a study by Lampignano et al. from 2021, they found that an index score developed to predict the risk of NAFLD was associated with a higher risk of all-cause dementia [90]. Another study that included 646 NAFLD cases did not find an association between NAFLD and a risk of dementia, however, when integrating fibrosis stage in a risk prediction model with conventional risk factors, they found an improved prediction of dementia risk [94]. In a study by Labenz et al., no association between all-cause dementia and NAFLD was found [92]. However, in a study by Shang et al., they found that NAFLD was associated with a hazard ratio (95% confidence interval) of 1.38 (1.10–1.72) for risk of developing all-cause dementia [93].

Thus, the evidence on the association between NAFLD and a risk of dementia is conflicting. Should there be a causal association, the proposed biological mechanisms remain speculative. Since NAFLD includes a significant inflammatory component, toxic substances are not removed efficiently by the liver. These substances, including metals, could influence brain function [95]. In addition, ABC transporters expressed in the liver could be affected in patients with NAFLD and these transporters have been proposed to play an important role in the development of dementia [96]. However, based on the current review the most plausible explanation for the association between NAFLD and a risk of dementia is the shared risk factors between NAFLD and dementia rather than NAFLD being a causal risk factor itself.

## 9. Physical Inactivity

### 9.1. Alzheimer’s Disease

In a meta-analysis of 19 studies by Kivimäki et al. that included approximately 40,000 individuals with a mean baseline age of 46 years and 15 years of follow-up, a higher risk of Alzheimer’s disease was found in those who were physically inactive when follow-up was shorter than 10 years but not when follow-up was longer than 10 years [97]. In a prospective study that included more than 117,000 individuals, Juul Rasmussen et al. found that there was no association between physical activity in leisure time and risk of Alzheimer’s disease [98].

### 9.2. Non-Alzheimer Dementia

In a study by Juul Rasmussen et al., low physical activity in leisure time was observed to be associated with a hazard ratio of 1.60 (95% confidence interval 1.40–1.83) for non-Alzheimer dementia compared with high physical activity in leisure time [98].

### 9.3. All-Cause Dementia

Most observational studies investigating the association between physical activity and a risk of all-cause dementia have found that physical activity early in life reduces the risk of developing all-cause dementia later in life [2,99]. In a study by Zotcheva et al. that included approximately 29,000 individuals aged 30–60 years and 25 years of follow-up, they observed a reduced risk of all-cause dementia in those who were physically active [100]. In the Whitehall study that included approximately 10,000 individuals with a follow-up of 28 years, the results were less clear [101]. A study that included 191 women with 44 years of follow-up showed a lower risk of developing all-cause dementia for women with a high baseline peak fitness compared with medium and low baseline peak fitness [102]. In a large meta-analysis by Kivimäki et al., a higher risk of all-cause dementia was found in those who were physically inactive when the follow-up was shorter than 10 years but not when the follow-up was longer than 10 years [97]. However, whether physical inactivity is a cause or a consequence of all-cause dementia is hard to determine when relying on observational data only [2]. A large meta-analysis of trials on exercise and risk of all-cause dementia that included 39 randomized clinical trials showed that physical exercise improved cognitive function [103]. Another meta-analysis of randomized clinical trials showed that exercise improved global cognition in individuals with mild cognitive impairment [104]. A meta-analysis that included approximately 3000 individuals found that there was no significant effect of exercise on onset of all-cause dementia, however a trend towards favoring exercise [105].

A recently published report from the World Health Organization (WHO) recommended physical activity to individuals with normal cognition to reduce the risk of cognitive decline [106]. In a double-blind randomized controlled trial called the Finnish Geriatric Intervention Study to Prevent Cognitive Impairment and Disability (FINGER), multidomain intervention in the prevention of all-cause dementia was investigated in 2654 individuals aged 60–77 [107]. The intervention included diet, exercise, cognitive training, and vascular risk factor monitoring; they found an improvement in cognition in the intervention group compared with the control group [107], and these findings were independent of baseline characteristics [108]. A possible role of physical activity in the development of all-cause dementia could be through its effect on hypertension [109], obesity [110], or both. In summary, physical inactivity seems to be a potentially modifiable risk factor for all-cause dementia.

## 10. Smoking

### 10.1. Alzheimer’s Disease

The evidence on the association between smoking and risk of dementia is unclear, however, it is a potentially important and modifiable risk factor [111]. In a longitudinal study by Choi et al. that included approximately 46,000 men aged 60 years or older, never smokers had a lower risk of Alzheimer’s disease [112]. In a Mendelian randomization study by Nordestgaard et al., genetically predicted smoking was associated with a higher risk of Alzheimer’s disease, but no association was found for observational data [113].

### 10.2. Non-Alzheimer Dementia

A study by Nordestgaard et al. found neither an observational nor a genetic association between cumulative smoking and the risk of non-Alzheimer dementia [113].

### 10.3. All-Cause Dementia

In a study by Choi et al., never smokers had a lower risk of all-cause dementia [112]. Quitting smoking was also associated with a lower risk of all-cause dementia [112]. One study, investigating the effect of second-hand smoking and decline in memory in women, found that longer exposure to second-hand smoking was associated with a greater decline in memory [114]. In a study by Nordestgaard et al., genetically predicted cumulative smoking was associated with a higher risk of all-cause dementia, but no clear association was seen observationally [113].

The fact that smokers are also at risk of premature death increases the probability that they die before they reach the age for developing dementia. This introduces a competing risk of death in the association between smoking and dementia, further complicating the interpretation of data [115]. A possible role for smoking in the development of dementia could be via the effect of smoking on the vessels of the brain through hypertension [116] and atherosclerosis [117]. Thus, smoking seems to be a potentially modifiable risk factor for dementia, but more evidence is needed to be able to establish causality.

## 11. Diet

### 11.1. Alzheimer’s Disease

In a large unpublished prospective study by Kjeldsen et al. that included 94,000 individuals, low adherence to Danish dietary guidelines was not found to be associated with risk of developing Alzheimer’s disease [118]. In a review from 2019 that included 56 articles, van den Brink et al. found that high adherence to the Mediterranean, Dietary Approaches to Stop Hypertension (DASH), or Mediterranean-DASH Intervention for Neurodegenerative Delay, was associated with a lower risk of Alzheimer’s disease [119].

### 11.2. Non-Alzheimer Dementia

In the study by Kjeldsen et al., they found that low adherence to Danish dietary guidelines was associated with a higher risk of developing non-Alzheimer dementia [118].

### 11.3. All-Cause Dementia

In a prospective study of 1410 individuals, 99 incident cases of all-cause dementia, and 66 incident cases of Alzheimer’s disease, Féart et al. found that high adherence to a Mediterranean diet was associated with slower cognitive decline measured using the Mini-Mental State Examination (MMSE), but not with incident all-cause dementia [120]. Some research suggests that a Mediterranean or similar diet has beneficial effects in the prevention of all-cause dementia. However, diet intervention is notoriously difficult, thus, making results hard to interpret. In a prospective cohort study that included 1068 individuals followed for 9 years, they found that high adherence to a diet very similar to the Mediterranean diet was associated with slower cognitive decline compared with low and intermediate adherence to the same diet [121]. In a prospective study of 6425 postmenopausal women aged 65 to 79 that investigated the association between different dietary patterns and risk of incident mild cognitive decline and probable all-cause dementia, they found that most of the dietary patterns were not associated with a significantly altered risk of either mild cognitive impairment or probable all-cause dementia [122]. However, high adherence to the DASH diet was associated with a lower risk of mild cognitive impairment, and the Healthy Eating Index-2010 was associated with a higher risk of probable all-cause dementia [122]. In a review from 2019, van den Brink et al. found that high adherence to the Mediterranean, DASH, or Mediterranean-DASH Intervention for Neurodegenerative Delay was associated with less cognitive decline [119]. A potential mechanism behind the association between a Mediterranean diet (or similar diet) and low risk of all-cause dementia could be through the known effect of a Mediterranean diet and risk of atherosclerosis [123,124]. In conclusion, diet seems to be a potential modifiable risk factor for all-cause dementia.

## 12. The Gut Microbiome

### Alzheimer’s Disease

Recently, there has been increased interest regarding a possible association between the gut microbiota and neurological disorders [125]. A study from 2016 that included 73 individuals with cognitive impairment with or without brain amyloidosis and 10 controls showed an association between proinflammatory gut microbiota and cognitive impairment/brain amyloidosis [126]. In a study from 2012 that included 54 Alzheimer’s disease patients, they found that patients infected with H. Pylori were more cognitively impaired than those not infected [127]. In a review from 2019, it was suggested that gut microbiota might be a modifiable risk factor for the development of Alzheimer’s disease [128]. Certain bacteria populations have been shown to influence intestinal barrier and insulin resistance [129,130], and some dietary supplements have been shown to alter the ratio of specific gut bacteria (Firmicutes/Bacteroidetes) [130]. However, the largest influence on the gut microbiota throughout life is very likely to be diet [131]. Hence, the most plausible explanation for an association between gut microbiota and a risk of dementia is that the gut microbiota is merely a reflection of the diet which, as described earlier, influences the risk of developing dementia, including Alzheimer’s disease [119].

## 13. Genetics

### 13.1. Alzheimer’s Disease

Early-onset Alzheimer’s disease is defined as the onset of Alzheimer’s disease before the age of 65 [5] and it accounts for approximately 10% of all Alzheimer’s disease cases [132]. It can be further subdivided in a familial, sporadic, and autosomal dominant form. Mutations in genes encoding presenilin 1 or 2 (*PSEN1* or *PSEN2*) and amyloid precursor protein (*APP*) have been linked to early-onset Alzheimer’s disease [5]. The apolipoprotein E gene (*APOE*), which is the most important genetic risk factor for late-onset Alzheimer’s disease, has also been shown to be associated with a risk of early-onset Alzheimer’s disease [133]. *APOE* has three common alleles, i.e., ɛ2, ɛ3, and ɛ4, and these alleles code for isoforms of the protein differing by the amino acids 112 and 158 [62,134]. The ɛ4 allele is associated with the highest risk of Alzheimer’s disease, and the ɛ2 allele is associated with the lowest risk of Alzheimer’s disease [59]. However, only 5–10% of patients with early-onset Alzheimer’s disease are carrying a pathogenic mutation in *APP*, *PSEN1*, *PSEN2,* or the common apolipoprotein E (*APOE*) ɛ4 allele (See Figure 4) [5].

In addition to the well-established genes involved in the pathogenesis of early-onset Alzheimer’s disease, recent studies have suggested several other genetic risk factors. A whole-genome sequencing study from 2015 found that the gene encoding the chemokine eotaxin-1 was associated with age at onset of Alzheimer’s disease [135]. In a Turkish family with Alzheimer’s disease, a mutation in *NOTCH3*, normally associated with cerebral autosomal dominant arteriopathy with subcortical infarcts and leukoencephalopathy (CADASIL) [136], was identified [137]. In addition, mutations in the sorting protein-related receptor gene (SORL1) have been found in early-onset Alzheimer’s disease patients [138]. In a review by Lardenoije et al., they concluded that epigenetics was also important in the pathogenesis of Alzheimer’s disease, even though the exact mechanism remains unknown [139].

The neuropathological hallmarks of Alzheimer’s disease are extracellular accumulation of amyloid plaques and intraneuronal accumulation of neurofibrillary tangles [21,24]. Although some studies have reported a larger pathological burden in early-onset Alzheimer’s disease compared with late-onset Alzheimer’s disease, the overall pathologic features are very similar [5]. Thus, the two diseases are hard to distinguish by any other criterion than age at onset [5].

Genome-wide association studies have identified risk variants for late-onset Alzheimer’s disease within more than 60 genetic loci [140,141,142,143]. These genetic loci are either involved in lipid metabolism, the immune system, or synaptic plasticity/synaptic cell functioning (see Figure 5) [144,145]. The most important genetic risk factor for late-onset Alzheimer’s disease is the ɛ4 allele of apolipoprotein E [59,141]. The *APOE* genotype affects the concentrations of several plasma lipids, lipoproteins, and apolipoproteins including apolipoprotein E, LDL cholesterol, triglycerides, HDL cholesterol, and total cholesterol [62]. In some studies, the ɛ4 allele has also been shown to be associated with a risk of ischemic heart disease [146]. In 2011, Hollingworth et al. discovered several new loci associated with risk of Alzheimer’s disease, including ATP-binding cassette transporter A7 gene (*ABCA7*), CD2-associated protein gene (*CD2AP*), ephrin type-A receptor 1 gene (*EPHA1*), membrane-spanning 4A gene (*MS4A*), and sialic acid binding Ig-like lectin 3 (*CD33*) [147].

In a large GWAS from 2013, Lambert et al. found that several additional common variants were associated with a risk of Alzheimer’s disease, including complement C3b/C4b receptor 1 gene (*CR1*), bridging integrator 1 gene (*BIN1*), clusterin gene (*CLU*), phosphatidylinositol-binding clathrin assembly protein gene (*PICALM*), and Ras and Rab interactor 3 gene (*RIN3*) [141]. The same year, the triggering receptor expressed on myeloid cells 2 (*TREM2*) was found to be associated with a risk of Alzheimer’s disease in two independent studies [148,149]. The association between variation in *CLU* and a risk of Alzheimer’s disease has been confirmed in a large, prospective study, but no association with lipid or apolipoprotein levels was found [150]. In a large, prospective cohort study, Juul Rasmussen et al. investigated the association between a genetic risk score created using *PICALM, BIN1, CD2AP*, and *RIN3* and a risk of Alzheimer’s disease and they found that the risk score was associated with a higher risk [151]. While *PICALM, BIN1, CD2AP*, and *RIN3* are involved in synaptic plasticity, *CR1, CD33, MS4A, TREM2,* and *EPHA1* are thought to play a role in the immune system, and *CLU* and *ABCA7* are genes involved in lipid metabolism [141,145]. However, another study in 950 hypertensive women found that genetic variation in clusterin was associated with levels of LDL cholesterol [152]. The association between *ABCA7* and Alzheimer’s disease has been confirmed in several studies, but no association with plasma lipid levels was found [153,154]. However, in another study from 2017 that included 118 Alzheimer’s disease cases and 120 controls, Li et al. found that the ABCA7 rs3764650 genotype was associated with LDL and total cholesterol concentrations [155].

Recently, other genes associated with a risk of Alzheimer’s disease have been discovered. In a paper from 2015, Nordestgaard et al. found that a loss-of-function mutation in ATP-binding cassette transporter A1 (*ABCA1*) was associated with a higher risk of Alzheimer’s disease, and also with plasma levels of HDL cholesterol, total cholesterol, apolipoprotein E, and apolipoprotein AI [61]. *ABCA1* was recently recognized as a risk gene in the, to date, largest GWAS of Alzheimer’s disease [143].

### 13.2. Non-Alzheimer Dementia

Until recently, most of the focus has been on finding genetic risk factors for Alzheimer’s disease. However, newer studies have also investigated genetic risk factors for other types of dementia including vascular dementia. In a study by Juul Rasmussen et al., the genetic risk score using *PICALM, BIN1, CD2AP*, and *RIN3* was also associated with a risk of vascular dementia [151]. A study that investigated the association between genetic variation in *CETP* and a risk of vascular endpoints, including ischemic heart disease, mortality, and dementia, found that the genetic variation associated with lower plasma LDL cholesterol-, lower triglyceride-, and higher HDL cholesterol concentrations was associated with a lower risk of ischemic heart disease and vascular dementia [63]. Genetic variation in *PCSK9* and *HMGCR* in combination was found to be associated with risk of vascular dementia in the study by Benn et al. [69]. These are genes that are both associated with a risk of ischemic heart disease [156,157].

## 14. The Vascular Hypothesis of Alzheimer’s Disease

In 1993, de la Torre and Mussivand proposed that the neurodegenerative changes seen in Alzheimer’s disease occurred downstream from the start of a chronic brain hypoperfusion [158]. Among the causes of hypoperfusion are hypertension that can induce atherosclerosis of extracranial and intracranial arteries and lead to microbleeds, white matter lesions, and microinfarcts [79]. Atherosclerosis can also cause hypoperfusion through ischemic infarcts. Another risk factor that contributes to the development of atherosclerosis is smoking [117] (see Figure 6). In support of this hypothesis, chronic hypertension and atherosclerosis have been shown to be risk factors for dementia, including Alzheimer’s disease and vascular dementia [159,160].

With age, blood flow to tissues including the brain diminishes [161]. When people reach the age of 65, blood flow may have declined 20% from what it was at age 20 [162]. In addition to this age-induced decline in cerebral blood flow (CBF), vascular risk factors could worsen this decline—a phenomenon named critically attained threshold of cerebral hypoperfusion (CATCH) by de la Torre et al. [163]. CATCH can be described as the beginning of CBF insufficiency resulting in an imbalance between neuronal demand and supply [158]. Over decades and depending on the existence of other risk factors for dementia including genetics, CATCH can compromise neuron/astroglial metabolism by limiting the delivery of nutrients to the brain. This causes an ischemic-hypoxic state at an early stage of cognitive impairment (see Figure 6) [164].

Beta amyloid is created by cleavage of amyloid precursor protein (APP) by β- and γ-secretases [23]. The γ-secretase complex is composed of four proteins: presenilin 1 or 2 (PSEN1 or PSEN2), nicastrin (NCT), anterior pharynx defective 1 (APH-1), and presenilin enhancer 2 (PEN2) [23]. Recent studies have suggested that these secretases were stimulated by hypoxia, and that hypoxia-inducible factor 1-alpha (HIF 1α) was involved in this activation [23]. When the brain is in a state of normoxia, the amyloid precursor protein is instead cleaved by the α-secretases causing it to be cleaved through the β-amyloid domain so that less β-amyloid is produced (see Figure 7) [23]. The APP/γ-secretase pathway seems to be involved in the control of the hypoxia response, and other proteins linked to Alzheimer’s disease pathology, for example, the Notch protein [23].

In gerbils (a form of small rodents), protein synthesis has been shown to decline at blood flows of <100 mL/100 g/min and approach zero at blood flows of 40 mL/100 g/min [165]. Studies in zebrafish have indicated that hypoxia induced the transcription of genes involved in the processing of amyloid precursor protein [23], suggesting that this mechanism has been preserved through evolution, and that the processing of amyloid precursor protein could be involved in functions that protect the brain against hypoxia [23]. Of relevance, recently, in the latest Alzheimer’s disease GWAS, the amyloid precursor protein gene (APP) has finally been found to be associated with a risk of late-onset Alzheimer’s disease [142,143]. In conclusion, an alternative hypothesis to the amyloid cascade hypothesis suggests that hypoxia is the underlying and primary pathologic event leading to the development of Alzheimer’s disease, and that the accumulation of β-amyloid might be a downstream consequence of cerebral hypoperfusion rather than the cause of Alzheimer’s disease [23].

## 15. Conclusions and Perspectives

The global dementia burden is expected to almost triple from approximately 46 million cases in 2015 to 132 million cases in 2050 [166], thus, making the human and economic costs of this disease an ever more important concern for societies around the world. Thus, updating the knowledge and understanding of dementia in future research is of utmost importance.

Accumulating evidence suggests that the overlap between risk factors for atherosclerotic cardiovascular disease and dementia is vast [2,16,18,101]. The differentiation between Alzheimer’s disease and vascular dementia originates from a time when only a clinical diagnosis was possible. Since the introduction of brain imaging techniques such as magnetic resonance imaging (MRI) and PET scans and cerebrospinal fluid examinations, it has become evident that there is no clear distinction between these two diseases [167,168,169,170,171]. Thus, it seems more reasonable to consider the two diseases as part of a spectrum of diseases instead of pure forms of either disease [170]. As suggested by Emrani et al., “rather than viewing Alzheimer’s disease and Vascular dementia as the two most common types of dementia, it may be more accurate to say that dementia-related proteinopathies with vascular disorders are the most common mechanisms underlying insidious onset dementia” [170].

Shared risk factors between atherosclerotic cardiovascular disease and dementia includes midlife obesity, dyslipidemia, midlife hypertension, diabetes, NAFLD, physical inactivity, smoking, diet, the gut microbiome composition, and genetics. For an overview of the associations between these risk factors and a risk of dementia see Figure 8. The pathogenic pathway from these risk factors to dementia is complex and the associations are often only shown observationally, making it difficult to draw conclusions about causality. Recent genetic studies aiming at elucidating these associations could contribute to further clarification of the pathogenic pathways and the question of causality. However, randomized clinical trials are needed for a final establishment of causality.

One theory suggests that hypoxia is the underlying main pathogenic event that leads, further on, to the development of dementia. Hypoxia can be caused by atherosclerosis or hemorrhages, which again can be caused by dyslipidemia, hypertension, diabetes, and smoking. Dyslipidemia, hypertension, and diabetes can all be caused by obesity that can be influenced by the level of physical activity and diet composition (See Figure 9). All these risk factors can be affected by genetics. Alzheimer’s disease could be a consequence of long-term hypoxia in the brain, whereas vascular dementia most likely is caused by more acutely occurring hypoxia caused by ischemic or hemorrhagic strokes.

Preventing dementia in the future is of utmost importance and incorporating modifiable cardiovascular risk factors to reach this goal is critical [2,172]. However, to be able to efficiently prevent any disease an in-depth understanding of the underlying pathogenesis is pivotal. Many years of research in dementia and particularly in Alzheimer’s disease have not resulted in any curative treatment. Perhaps because research in curative treatments has been guided by a questionable hypothesis. Rethinking the causes of dementia might pave the way for better means of preventing these devastating diseases [171]. Furthermore, broadening the perspective of the understanding of the underlying mechanisms behind the development of dementia might prevent large expenditures and efforts put into futile research and drug development. Efforts should, instead, be focused on and guided by recent science [19]. This would not only save societies from the social and economic burden of an ageing demented population, but also help individuals in making lifestyle choices that will enhance the possibility of a healthy and active senescence.

## Figures and Tables

**Figure 1 ijms-23-09777-f001:**
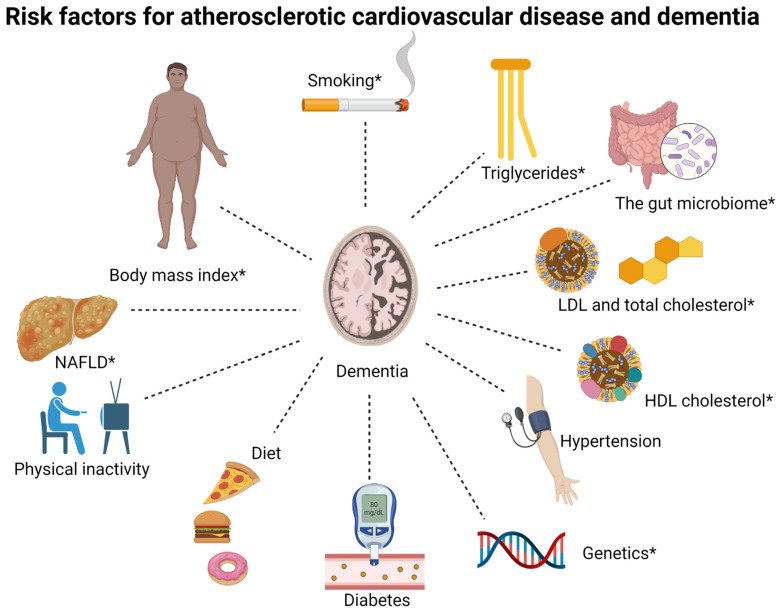
**Overview of shared risk factors between dementia and atherosclerotic cardiovascular disease, discussed in this review.** LDL, low-density lipoprotein; HDL, high-density lipoprotein; NAFLD, non-alcoholic fatty liver disease. The associations for some risk factors differ between Alzheimer’s disease and non-Alzheimer dementia, specifically, body mass index, triglycerides, HDL cholesterol, NAFLD, the gut microbiome, smoking, and genetics (these are marked by *).

**Figure 2 ijms-23-09777-f002:**
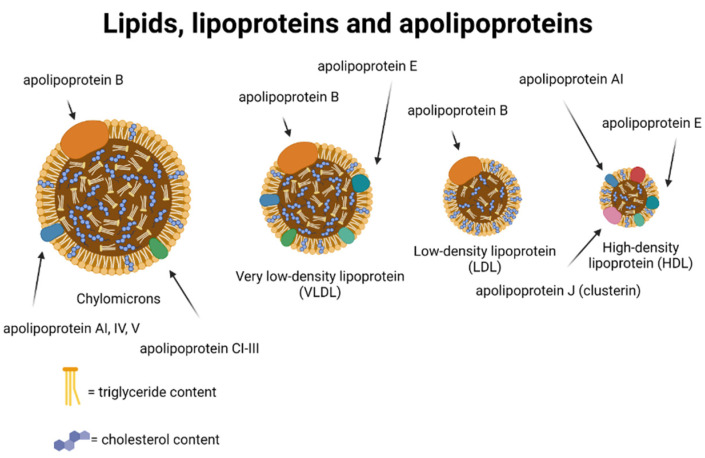
**Illustration of plasma lipids, lipoproteins, and apolipoproteins.** Note, only the apolipoproteins most relevant for this review are shown. Created with Biorender.com.

**Figure 3 ijms-23-09777-f003:**
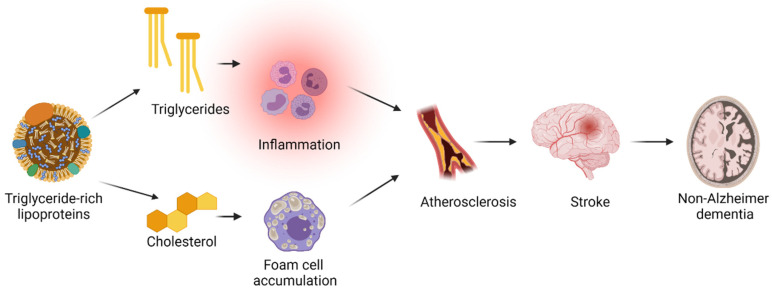
**A proposed explanation for the association between high plasma triglycerides and a high risk of stroke and non-Alzheimer dementia.** Triglyceride concentration reflects the triglyceride content in triglyceride-rich lipoprotein particles, and thus, also the cholesterol content in these particles. The cholesterol content can be taken up by macrophages leading to foam cell formation. These foam cells get trapped in the arterial wall and lead to the formation of an atherosclerotic plaque. The triglyceride content of triglyceride-rich lipoprotein particles can cause inflammation, and thus, further contributing to the formation of atherosclerosis. Atherosclerosis causes ischemic stroke which, again, causes non-Alzheimer dementia. Created with BioRender.com.

**Figure 4 ijms-23-09777-f004:**
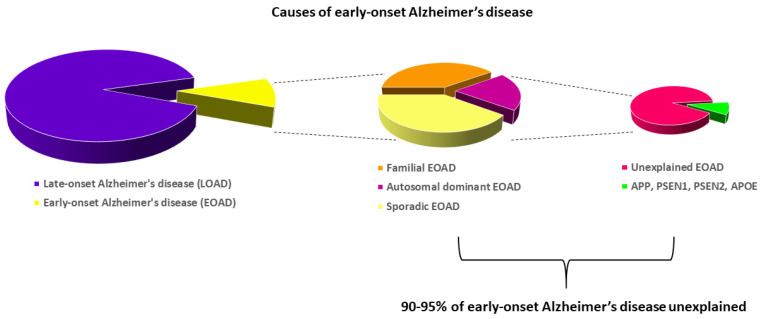
Causes of early-onset Alzheimer’s disease. Early-onset Alzheimer’s disease (EOAD) accounts for approximately 10% of Alzheimer’s disease cases. Mutations in the amyloid precursor protein gene (*APP*), the presenilin 1 and 2 genes (*PSEN1* and *-2*), and the apolipoprotein E gene (*APOE*) explain only a small part of autosomal dominant early-onset Alzheimer’s disease, leaving large parts of early-onset Alzheimer’s disease unexplained. Adapted from Cacace et al., Alzheimer’s Dement 2016.

**Figure 5 ijms-23-09777-f005:**
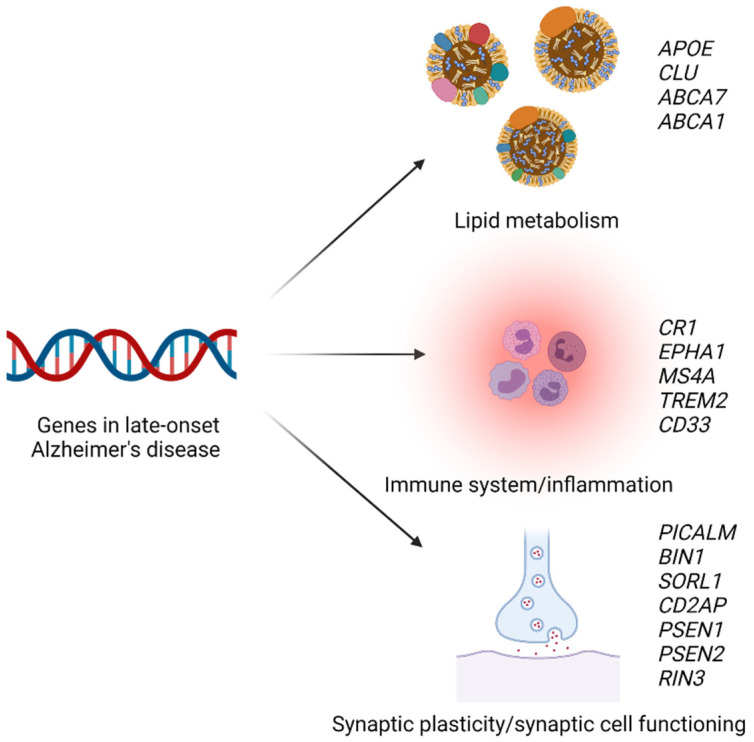
**Selected genetic risk factors for late-onset Alzheimer’s disease.** Genes associated with a risk of Alzheimer’s disease are involved in lipid metabolism, the immune system and inflammation, or synaptic plasticity and synaptic cell functioning. APOE, apolipoprotein E gene; CLU, clusterin gene; ABCA7, ATP-binding cassette transporter A7 gene; ABCA1, ATP-binding cassette transporter A1 gene; CR1, complement C3b/C4b receptor 1 gene; EPHA1, ephrin type-A receptor 1 gene; MS4A, membrane-spanning 4A gene; TREM2, triggering receptor expressed on myeloid cells 2 gene; PICALM, phosphatidylinositol binding clathrin assembly protein gene; BIN1, bridging integrator 1 gene; SORL1, sorting protein-related receptor gene; CDA2P, CD2-associated protein gene; PSEN1/2, presenilin 1 or 2 gene; RIN3, Ras and Rab interactor 3 gene; CD33, sialic acid binding Ig-like lectin 3 gene. Created with Biorender.com.

**Figure 6 ijms-23-09777-f006:**
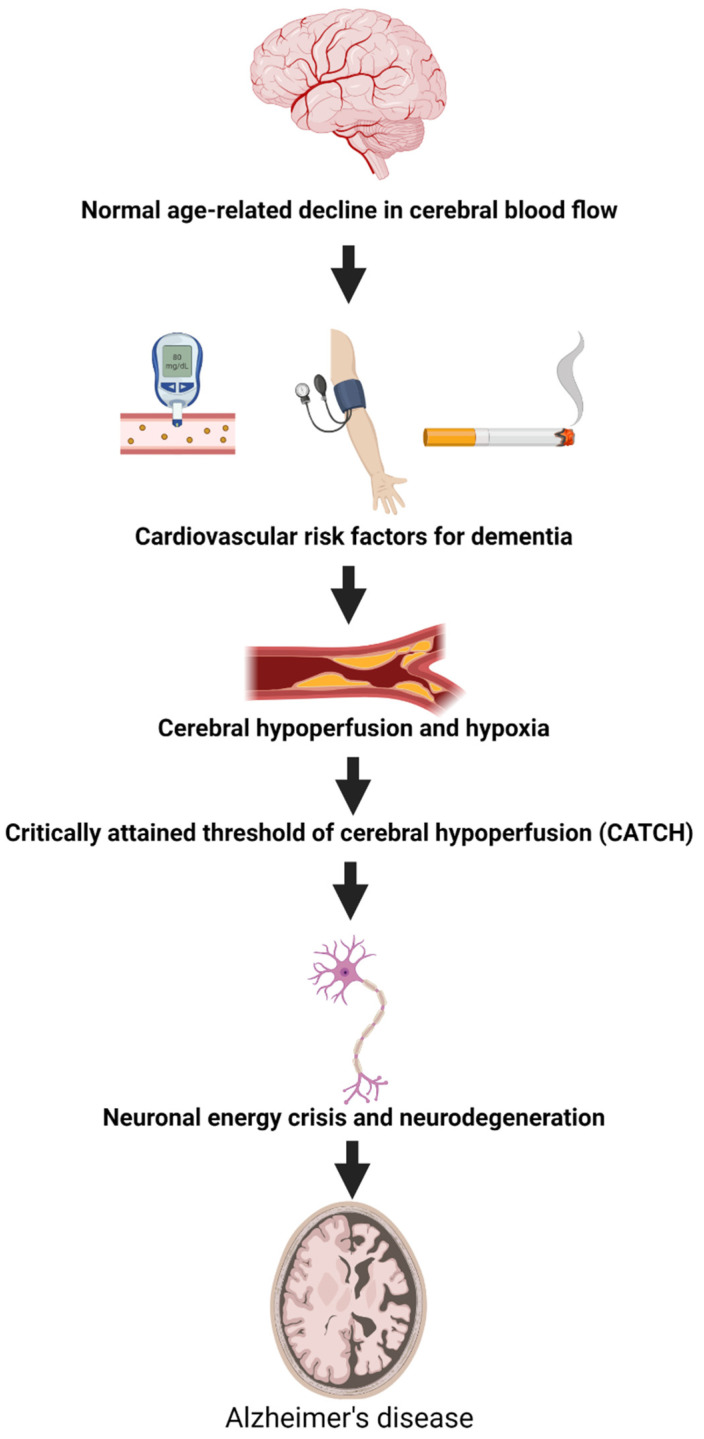
**The vascular hypothesis of Alzheimer’s disease.** Normal ageing causes a decline in cerebral blood flow (1). The existence of cardiovascular risk factors such as smoking, high blood pressure, and diabetes will worsen this decline (2) leading to cerebral hypoperfusion and hypoxia (3). At one point a critical threshold of cerebral hypoperfusion is reached (4) leading to neuronal energy crisis and neurodegeneration (5), eventually causing Alzheimer’s disease (6). Adapted from de la Torre Journal of Alzheimer’s disease 2018 [158].

**Figure 7 ijms-23-09777-f007:**
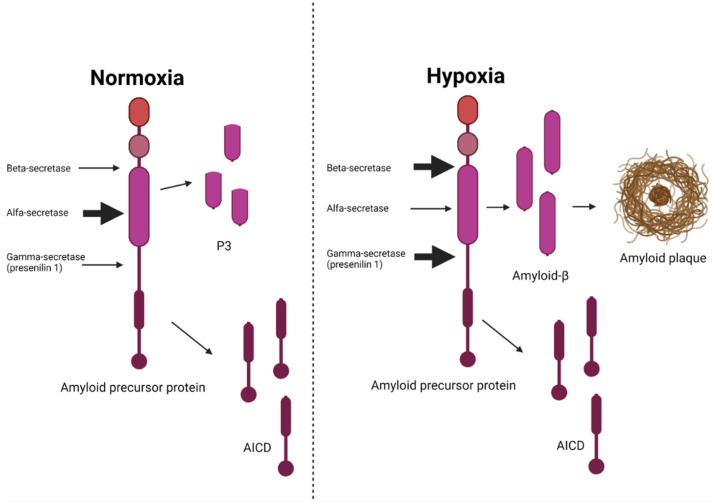
**The impact of hypoxia/ischemia on the processing of amyloid precursor protein.** During normoxia, the alfa-secretase is more active than the beta- and gamma-secretases resulting in very little production of β-amyloid. During hypoxia, the beta- and gamma-secretases are stimulated causing an excess production of β-amyloid leading to the formation of amyloid plaques—a pathological hallmark of Alzheimer’s disease. AICD, amyloid precursor protein intracellular domain. Adapted from Salminen et al., J. Neurochem 2017 [23]. Created with Biorender.com.

**Figure 8 ijms-23-09777-f008:**
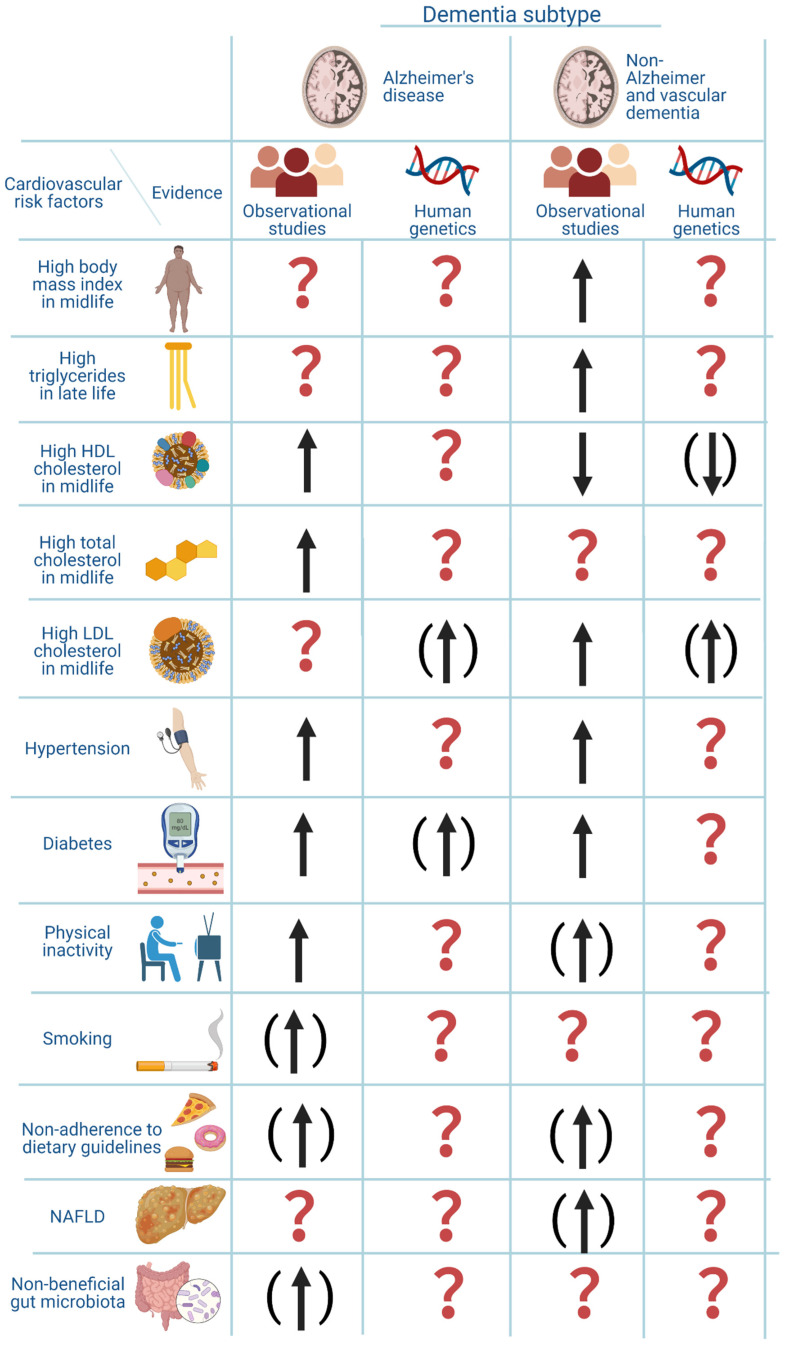
**Overview of atherosclerotic cardiovascular risk factors and their association with risk of dementia**. The figure is based on the current review of the literature. The arrows indicate the direction of the association. An arrow without parenthesis means moderate to good evidence. An arrow with parenthesis means weak evidence. A question mark means that the evidence is not available or very unclear. The genetic evidence for LDL cholesterol was based on the article by Benn et al. that investigated the association between low LDL cholesterol and dementia, and the association was assumed to be similar but in the opposite direction for high LDL cholesterol and risk of dementia. Created with Biorender.com.

**Figure 9 ijms-23-09777-f009:**
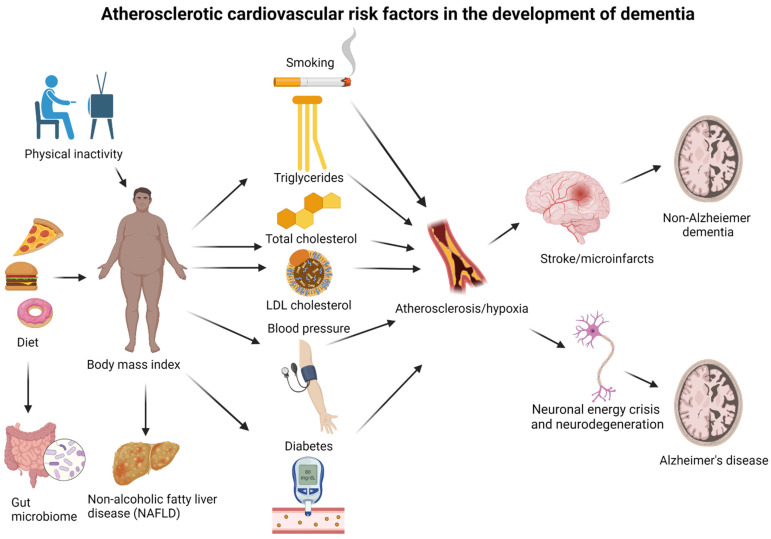
**Proposed pathways from atherosclerotic cardiovascular risk factors for dementia to the development of disease.** (1) The body mass index is influenced by the level of physical activity and diet, and the gut microbiome is influenced by diet composition. (2) The body mass index influences LDL cholesterol and triglyceride concentrations, blood pressure, and risk of diabetes and NAFLD. (3) Smoking, LDL cholesterol, triglycerides, blood pressure, and diabetes are all risk factors for developing atherosclerosis. (4) Atherosclerosis can cause long-term reduced cerebral blood flow and hypoxia if vessels are not completely blocked, leading to neuronal energy crisis, neurodegeneration, and eventually, Alzheimer’s disease. Atherosclerosis can also cause acute hypoxia due to strokes, leading to vascular dementia. The association between high levels of HDL cholesterol and a high risk of dementia might be due to reverse causation caused by a low body mass index or due to high alcohol consumption. All risk factors can be influenced by genetics. Created with Biorender.com.

## Data Availability

Not applicable.

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
