# Peer review of "Shared Risk Factors between Dementia and Atherosclerotic Cardiovascular Disease"

_ijms, 2022, doi:10.3390/ijms23179777_

Round 1

Reviewer 1 Report

The manuscript reviews the cardiovascular risk factors of dementia. However, it doesn’t add anything more than previous reviews (such as Livingston et al review: the second reference) to the field. If the review could compare the contribution of cardiovascular risk factors in Alzheimer’s disease vs. cardiovascular dementia, it would add something to the literature.

Besides the lack of novelty, the text is not comprehensive enough:

1.     Dementia, Alzheimer’s, and vascular dementia are the concepts that the authors talked about in the paper, but not in a systematic way. In each paragraph there is a mixture of evidence for each of these concepts. Thus, it is not completely clear if the focus of review is dementia in general, or Alzheimer’s disease and vascular dementia. Additionally, sometimes the statement is about dementia but supporting references are for Alzheimer’s disease (line 34).

2.     “… an alternative hypothesis for the pathogenesis behind the development of Alzheimer’s disease will be discussed” was considered as one of the aims of study, while this cannot be an aim of this review. It can be a conclusion.

3.     There is a lack of attention to details that is necessary for a concise scientific writing. For example, HDL cholesterol and physical activity have been considered as risk factors throughout the text, while they are protective factors.

Author Response

Dr. Liko Wang

Assigned editor, International Journal of Molecular Sciences

Special issue, “Impact of Lipid Homeostasis in Neurodegenerative Diseases

Copenhagen, August 19th, 2022

Dear Dr. Liko Wang,

We are most thankful for the opportunity to revise our review entitled “Cardiovascular risk factors for dementia” for the special issue of International Journal of Molecular Sciences “Impact of Lipid Homeostasis in Neurodegenerative Diseases”. To specify the fact that the risk factors we are discussing are risk factors specifically for atherosclerotic cardiovascular disease we have changed our title to “Shared risk factors between dementia and atherosclerotic cardiovascular disease”. If either you or the reviewers find this title inappropriate, we are happy to change it back again.

We have done our utmost to accommodate the reviewer’s comments (for point-by-point response see below). Throughout the review we have now separated each paragraph on a certain risk factor in subsections for each of the three categories of dementia (Alzheimer’s disease, Non-Alzheimer dementia, and all-cause dementia), that we are describing. This has been done to accommodate reviewer number one and has been done to the extent it has been possible based on the currently available science. We have also changed the risk factor “physical activity” to “physical inactivity” as requested by reviewer number one.

Further, to accommodate reviewer number two we have included two extra subsections. One subsections discussion the role of NAFLD as a risk factor for dementia, and one subsection discussing the role of the gut microbiome as a risk factor for dementia. In these subsections we are also referring to the articles requested by the reviewer. Finally, a few sections have been removed since we did not think that the contributed to the overall aim of the review.

We hope that you find the revised version of our review suitable for publication in International Journal of Molecular Sciences.

Sincerely yours,

Liv Tybjærg Nordestgaard MD, PhD

Department of Clinical Biochemistry

Copenhagen University Hospital - Rigshospitalet

Blegdamsvej 9, DK-2100 Copenhagen, Denmark

Phone: +45 3545 3946

E-mail: liv.tybjaerg.nordestgaard@regionh.dk

Reviewer one: Comments and Suggestions for Authors

The manuscript reviews the cardiovascular risk factors of dementia. However, it doesn’t add anything more than previous reviews (such as Livingston et al review: the second reference) to the field. If the review could compare the contribution of cardiovascular risk factors in Alzheimer’s disease vs. cardiovascular dementia, it would add something to the literature.

Response: Thank you very much for taking the time to read our review and thank you for your constructive comments. We have now clarified throughout the review when we are discussing risk factors for Alzheimer’s disease and when we are discussing risk factors for non-Alzheimer dementia (including vascular dementia and unspecified dementia). Further, we have incorporated in the legend to figure 1 the differences between risk factors for Alzheimer’s disease and Non-Alzheimer dementia.

Besides the lack of novelty, the text is not comprehensive enough:

  1. Dementia, Alzheimer’s, and vascular dementia are the concepts that the authors talked about in the paper, but not in a systematic way. In each paragraph there is a mixture of evidence for each of these concepts. Thus, it is not completely clear if the focus of review is dementia in general, or Alzheimer’s disease and vascular dementia. Additionally, sometimes the statement is about dementia but supporting references are for Alzheimer’s disease (line 34).

Response: Thank you very much for this constructive comment. We have now done our utmost to clarify when we are describing risk factors for Alzheimer’s disease, Non-Alzheimer dementia (including vascular dementia), and all-cause dementia throughout the text. We are sorry about the mistake in line 34 which has now been corrected.

  1. “… an alternative hypothesis for the pathogenesis behind the development of Alzheimer’s disease will be discussed” was considered as one of the aims of study, while this cannot be an aim of this review. It can be a conclusion.

Response: Thank you for mentioning this. We have now removed the sentence from the abstract and the introduction of the review.

  1. There is a lack of attention to details that is necessary for a concise scientific writing. For example, HDL cholesterol and physical activity have been considered as risk factors throughout the text, while they are protective factors.

Response: We apologize for this inconsistency. We have now changed physical activity to physical inactivity throughout the review. HDL cholesterol on the other hand is more complex as a risk factor for dementia. For example, is high HDL cholesterol associated with high risk of developing Alzheimer’s disease and all-cause dementia (see Kjeldsen et al. Plasma High-Density Lipoprotein Cholesterol and Risk of Dementia: Observational and Genetic Studies. Cardiovasc. Res. 2021, doi:10.1093/cvr/cvab164). For this reason we have not changed the mentioning of HDL cholesterol.For diet we have changed the symbols from healthy to unhealthy foods to imply that an unhealthy diet is the risk factor for developing dementia.

Reviewer 2 Report

The authors developed a Review regarding the Cardiovascular risk factors for dementia. Very suggestive graphical part. I have few suggestions, as follows:

Remove the empty spaces between the paragraphs. The aspect will be more professional.

Please make 2 separate subsections where you discuss:

-   The correlation between NAFLD and the risk of Alzheimer disease, being known that NAFLD is a significant inflammatory condition, toxic substances are not removed efficiently by the liver, and they influence brain function. I suggest checking and referring to https://doi.org/10.1007/s12035-020-02096-w ;

https://doi.org/10.1007/s12035-020-02211-x

- The other chapter should analyse the role of gut microbiota in Alzheimer given the fact that certain bacteria population influence intestinal barrier and produce insulin-resistance, pleas detail the potential role of Bacteroides, Firmicutes species. Please check and refer to https://doi.org/10.3390/microorganisms9030618

Some references are very old (more than 30 years). Maybe you can update those references, as news about Alzheimer is abundant in the literature.

Author Response

Dr. Liko Wang

Assigned editor, International Journal of Molecular Sciences

Special issue, “Impact of Lipid Homeostasis in Neurodegenerative Diseases

Copenhagen, August 19th, 2022

Dear Dr. Liko Wang,

We are most thankful for the opportunity to revise our review entitled “Cardiovascular risk factors for dementia” for the special issue of International Journal of Molecular Sciences “Impact of Lipid Homeostasis in Neurodegenerative Diseases”. To specify the fact that the risk factors we are discussing are risk factors specifically for atherosclerotic cardiovascular disease we have changed our title to “Shared risk factors between dementia and atherosclerotic cardiovascular disease”. If either you or the reviewers find this title inappropriate, we are happy to change it back again.

We have done our utmost to accommodate the reviewer’s comments (for point-by-point response see below). Throughout the review we have now separated each paragraph on a certain risk factor in subsections for each of the three categories of dementia (Alzheimer’s disease, Non-Alzheimer dementia, and all-cause dementia), that we are describing. This has been done to accommodate reviewer number one and has been done to the extent it has been possible based on the currently available science. We have also changed the risk factor “physical activity” to “physical inactivity” as requested by reviewer number one.

Further, to accommodate reviewer number two we have included two extra subsections. One subsections discussion the role of NAFLD as a risk factor for dementia, and one subsection discussing the role of the gut microbiome as a risk factor for dementia. In these subsections we are also referring to the articles requested by the reviewer. Finally, a few sections have been removed since we did not think that the contributed to the overall aim of the review.

We hope that you find the revised version of our review suitable for publication in International Journal of Molecular Sciences.

Sincerely yours,

Liv Tybjærg Nordestgaard MD, PhD

Department of Clinical Biochemistry

Copenhagen University Hospital - Rigshospitalet

Blegdamsvej 9, DK-2100 Copenhagen, Denmark

Phone: +45 3545 3946

E-mail: liv.tybjaerg.nordestgaard@regionh.dk

Reviewer two: Comments and Suggestions for Authors

The authors developed a Review regarding the Cardiovascular risk factors for dementia. Very suggestive graphical part. I have few suggestions, as follows:

Remove the empty spaces between the paragraphs. The aspect will be more professional.

Response: Thank you very much for taking the time to read our review and for your constructive comments. Unfortunately, given the article template that is provided by the journal it is not possible to remove all empty spaces between paragraphs at the moment. We have removed empty spaces as far as possible and will see to that this will be done completely when proofreading the manuscript.

Please make 2 separate subsections where you discuss:

-   The correlation between NAFLD and the risk of Alzheimer disease, being known that NAFLD is a significant inflammatory condition, toxic substances are not removed efficiently by the liver, and they influence brain function. I suggest checking and referring to https://doi.org/10.1007/s12035-020-02096-w ;

https://doi.org/10.1007/s12035-020-02211-x. 

Response: Thank you very much for this suggestion. We have now incorporated 2 separate subsections where we discuss 1) NAFLD and the risk of Alzheimer disease, and 2) the role of gut microbiota in Alzheimer disease.

Thank you for the suggestions of these articles. We are now referring to them in the subsection on NAFLD and Alzheimer disease.

- The other chapter should analyse the role of gut microbiota in Alzheimer given the fact that certain bacteria population influence intestinal barrier and produce insulin-resistance, pleas detail the potential role of Bacteroides, Firmicutes species. Please check and refer to https://doi.org/10.3390/microorganisms9030618

Response: Thank you for the suggestions of this article. We are now referring to it in the subsection on the gut microbiota.

Some references are very old (more than 30 years). Maybe you can update those references, as news about Alzheimer is abundant in the literature.

Response: Thank you very much for this comment. We have now included three new references:

  1. Nordestgaard, A.T.; Nordestgaard, B.G.; Frikke-Schmidt, R.; Juul Rasmussen, I.; Bojesen, S.E. Self-Reported and Genetically Predicted Coffee Consumption and Smoking in Dementia: A Mendelian Randomization Study. Atherosclerosis 2022, 348, 36–43, doi:10.1016/j.atherosclerosis.2022.03.022.
  2. Rasmussen, I.J.; Rasmussen, K.L.; Thomassen, J.Q.; Schnohr, P.; Frikke-schmidt, R.; General, C.; Study, P.; Hospital, G.; Copenhagen, T.; Heart, C.; et al. Physical Activity in Leisure Time and at Work and Risk of Dementia - a Prospective Cohort Study of 117 , 616 Individuals. 2022, 1–25.
  3. Kjeldsen, E.W.; Thomassen, J.Q.; Rasmussen, K.L.; Tybjærg-Hansen, A.; Nordestgaard, B.G.; Frikke-Schmidt, R. Adherence to Dietary Guidelines and Risk of Dementia: A Prospective Cohort Study of 94,184 Individuals. Epidemiol. Psychiatr. Sci.

Round 2

Reviewer 2 Report

All the requirements have been fulfilled. I recommend publication.